# The Role of RodA-Conserved Cysteine Residues in the *Aspergillus fumigatus* Conidial Surface Organization

**DOI:** 10.3390/jof6030151

**Published:** 2020-08-26

**Authors:** Isabel Valsecchi, Emmanuel Stephen-Victor, Sarah Sze Wah Wong, Anupama Karnam, Margaret Sunde, J. Iñaki Guijarro, Borja Rodríguez de Francisco, Thomas Krüger, Olaf Kniemeyer, Gordon D. Brown, Janet A. Willment, Jean-Paul Latgé, Axel A. Brakhage, Jagadeesh Bayry, Vishukumar Aimanianda

**Affiliations:** 1Aspergillus Unit, Institut Pasteur, 75015 Paris, France; valsecchiisabel@gmail.com (I.V.); jean-paul.latge@pasteur.fr (J.-P.L.); 2Institut National de la Santé et de la Recherche Médicale, Centre de Recherché des Cordeliers, Sorbonne Université, Université de Paris, 75006 Paris, France; esvkai@gmail.com (E.S.-V.); suma.anupama@gmail.com (A.K.); 3Molecular Mycology Unit, Institut Pasteur, CNRS-UMR-2000, 10098 Paris, France; swong@pasteur.fr; 4School of Medical Sciences and Sydney Nano, University of Sydney, Sydney, New South Wales (NSW) 2006, Australia; margaret.sunde@sydney.edu.au; 5Biological NMR Technological Platform, Institut Pasteur, CNRS-UMR-3528, 75015 Paris, France; inaki.guijarro@pasteur.fr (J.I.G.); brodrigu@pasteur.fr (B.R.d.F.); 6Department of Molecular and Applied Microbiology, Leibniz Institute for Natural Product Research and Infection Biology, 07745 Jena, Germany; thomas.krueger@hki-jena.de (T.K.);olaf.kniemeyer@hki-jena.de (O.K.); axel.brakhage@hki-jena.de (A.A.B.); 7Medical Research Council Centre for Medical Mycology (MRC CMM), University of Exeter, Exeter EX4 4QD, UK; gordon.brown@exeter.ac.uk (G.D.B.); J.Willment@exeter.ac.uk (J.A.W.); 8Institute of Microbiology, Friedrich Schiller University, 07745 Jena, Germany

**Keywords:** *Aspergillus**fumigatus*, host–fungal interaction, conidial surface, RodAp, hydrophobin, disulfide bonds, immunomodulation

## Abstract

Immune inertness of *Aspergillus*
*fumigatus* conidia is attributed to its surface rodlet-layer made up of RodAp, characterized by eight conserved cysteine residues forming four disulfide bonds. Earlier, we showed that the conserved cysteine residue point (ccrp) mutations result in conidia devoid of the rodlet layer. Here, we extended our study comparing the surface organization and immunoreactivity of conidia carrying ccrp-mutations with the *RODA* deletion mutant (∆*rodA*). Western blot analysis using anti-RodAp antibodies indicated the absence of RodAp in the cytoplasm of ccrp*-*mutant conidia. Immunolabeling revealed differential reactivity to conidial surface glucans, the ccrp-mutant conidia preferentially binding to α-(1,3)-glucan, ∆*rodA* conidia selectively bound to β-(1,3)-glucan; the parental strain conidia showed negative labeling. However, permeability of ccrp-mutants and ∆*rodA* was similar to the parental strain conidia. Proteomic analyses of the conidial surface exposed proteins of the ccrp-mutants showed more similarities with the parental strain, but were significantly different from the ∆*rodA*. Ccrp-mutant conidia were less immunostimulatory compared to ∆*rodA* conidia. Our data suggest that (i) the conserved cysteine residues are essential for the trafficking of RodAp and the organization of the rodlet layer on the conidial surface, and (ii) targeted point mutation could be an alternative approach to study the role of fungal cell-wall genes in host–fungal interaction.

## 1. Introduction

Even though conidia (the asexual spores) produced by *Aspergillus fumigatus*, an opportunistic airborne fungal pathogen, are constantly inhaled by humans, the conidial surface rodlet layer masks immediate immune recognition of the conidia, resulting in a balanced immune response [1,2]. The conidial germination is associated with controlled exposure of polysaccharides that are major components of the conidial inner cell wall and constitute pathogen-associated molecular patterns [3,4,5]. The outer coat of conidia consists of a rodlet layer composed of RodAp, a protein belonging to the Class-I hydrophobin family [6] that is characterized by the presence of eight conserved cysteine residues [7,8]. These cysteine residues are involved in the formation of four disulfide linkages that are critical in the structural organization of RodAp. Previously, we showed that the conserved cysteine residue point (ccrp) mutations in the RodA resulted in conidia devoid of a surface rodlet layer [9]. Indeed, ccrp-mutations of either single or of four cysteine residues that disrupt one or four disulfide linkages, respectively, resulted in mutant conidia with similar cell surface organization.

Here, we extend our study in understanding the consequences of ccrp-mutations in RodAp. We show in vitro by nuclear magnetic resonance (NMR) analysis that the formation of the disulfides bonds was necessary for the folding of RodAp into a monomer that is competent for rodlet formation. The Western blot of the single and quadruple ccrp-mutant conidial cytoplasmic contents probed with anti-RodAp antibodies indicated the absence of RodAp. Immunolabeling of these ccrp-mutant conidia was strongly positive for α-(1,3)-glucan and only weakly positive for β-(1,3)-glucan. In contrast, Δ*rodA* mutant conidia were strongly positive for β-(1,3)-glucan and only weakly positive for α-(1,3)-glucan. Proteomic analyses of the surface proteins after trypsin-shaving of these ccrp-mutant conidia showed similarities with parental strain conidia, but significant differences with the Δ*rodA* conidial surface proteome. Functional assays indicated that both single and quadruple ccrp-mutant conidia were less stimulatory than ∆*rodA* conidia. Overall, the surface characteristics of ccrp-mutant conidia showed similarities with swollen conidia, the first germination morphotype of *A*. *fumigatus*.

## 2. Materials and Methods 

Fungal strains and generation of RodA-conserved cysteine residue (s) point-mutants: The parental *A*. *fumigatus* strain used in this study was CEA17_Δ*akuB*^KU80^. The RodAp-cysteine residue point-mutant strains and the *RODA* deletion mutant were generated as described earlier [9,10,11]. All the strains were maintained on 2% malt–agar slants at ambient temperature. Conidia were harvested from 10–12-days old malt–agar culture slants and washed twice using 0.05% aqueous Tween-20. Immunolabeling and interaction studies with human dendritic cells (DCs) were performed with *p*-formaldehyde (PFA) fixed conidia that were obtained as described earlier [1].

Conidial permeability analysis and melanin extraction: Conidia (unfixed, 10^6^) were incubated with Calcofluor-white (5 µg/mL; 50 µL) for 5 min at ambient temperature, washed thrice with 0.05% aqueous Tween-20, mounted on slides and observed under a fluorescence microscope (EVOS FL Cell Imaging System, Life Technologies, Thermo-Fisher Scientific, Illkirch CEDEX, France). Fluorescein isothiocyanate (FITC) labeling was performed by incubating conidia with FITC in 50 mM sodium carbonate buffer (pH 9.0) for 30 min in the dark at ambient temperature, washed thrice with carbonate buffer and observed under a fluorescence microscope after mounting on the slides. Melanin pigment extraction from the parental strain and mutant conidia were performed following the protocol as described earlier [12]. Briefly, conidia were treated with a proteinase-K (Sigma-Aldrich, Saint-Quentin-Fallavier, France) and Glucanex (Novozymes, Dittingen, Switzerland; glycohydrolytic enzyme) mixture in 1.0 M sorbitol–0.1 M sodium citrate buffer, pH 5.5 (overnight at 37 °C), followed by denaturation (4.0 M guanidine thiocyanate, 10–12 h at ambient temperature) and incubation with 6.0 M HCl (100 °C, 1 h). The dark particles (melanin ghosts) obtained were dialyzed against water for 8–10 days and freeze-dried. The parental strain conidia (10^10^ conidia) yielded 1.80 ± 0.15 mg of melanin pigment (in dry weight); the changes in the dry weight of the melanin pigments extracted from the mutant conidia were calculated accordingly.

Conidial surface labeling for different cell wall/surface components: Fixed conidia (10^6^) were incubated with 50 µL of wheat germ agglutinin-FITC (WGA-FITC) or Concanavalin-A (ConA)-FITC (5 µg/mL) in 50 mM sodium carbonate buffer (pH 9.0) for 30 min in the dark at ambient temperature, for chitin and mannan labeling on the conidial surface, respectively. After washing thrice with the carbonate buffer, conidia were observed under a fluorescence microscope. Melanin labelling was performed using MelLec, the melanin-binding receptor, as described earlier [13]. For β-(1,3)-glucan labeling, human Fc-conjugated Dectin-1 (5 µg/mL) in PBS containing bovine serum albumin (1%) (PBS-BSA) and for α-(1,3)-glucan labeling, mouse polyclonal antibody [14] diluted in PBS-BSA (1:200 dilution), were added to fixed conidia (10^6^), incubated at ambient temperature for 1 h, washed thrice with PBS, incubated at ambient temperature with human Fc-specific IgG-FITC (β-(1,3)-glucan) or anti-mouse IgG-TRITC/FITC (α-(1,3)-glucan), washed thrice with PBS, resuspended in Mowiol embedding medium, mounted on slides and subjected to fluorescence microscopy. Chemical removal of the conidial surface rodlet layer using 48% hydrofluoric acid (HF) was performed as described earlier [1], but conidia were incubated in HF for 20 h at 4 °C; HF treated conidia were washed with water at least five times before taking conidia for immunolabeling. An aliquot of labeled conidia was also taken for flow-cytometric analysis; the samples were acquired by using LSR II (BD Biosciences, Le Pont de Claix, France), and the data were analyzed by BD FACS DIVA v8.0.1 (BD Biosciences, Le Pont de Claix, France) and FlowJo (FlowJo, LLC, Ashland, KY, USA).

Extraction of conidial surface proteins by trypsin shaving: Surface proteins were extracted as performed previously [15,16]. Briefly, *A*. *fumigatus* were grown for seven days at 37 °C on *Aspergillus* minimal medium (70 mM NaNO_3_, 11.2 mM KH_2_PO_4_, 7 mM KCl, 2 mM MgSO_4_, and 1 µL/mL trace element solution (pH 6.5)) agar plates with 1% (*w*/*v*) glucose. The trace element solution was composed of 1 g FeSO_4_ • 7 H_2_O, 8.8 g ZnSO_4_ • 7 H_2_O, 0.4 g CuSO_4_ • 5 H_2_O, 0.15 g MnSO_4_ • H_2_O, 0.1 g NaB_4_O_7_ • 10 H_2_O, 0.05 g (NH_4_)_6_Mo_7_O_24_ • 4 H_2_O, and ultra-filtrated water to 1000 mL [17,18]. Conidia collected from three 10 cm plates (~10^9^ conidia) were used immediately after collection in water, washed thrice with 25 mM ammonium bicarbonate, and collected by centrifugation (1800× *g*, 10 min, ambient temperature). The conidia were resuspended in 800 µL of 25 mM ammonium bicarbonate and treated with 5 µg MS-approved trypsin (Serva) for 5 min at 37 °C with gentle agitation. Samples were then immediately passed through a 0.2 µm cellulose acetate filter (Sartorius, Gottingen, Germany) and collected in a microcentrifuge tube, followed by washing of the syringe filter with 200 µL of 25 mM ammonium bicarbonate; 9 µL of 89% (*v*/*v*) formic acid was added to stop the tryptic digestion and the samples were dried using a SpeedVac Concentrator (Thermo-Fisher Scientific, Waltham, MA, USA), resuspended in 25 µL of 2% (*v*/*v*) acetonitrile and 0.05% (*v*/*v*) trifluoroacetic acid, and centrifuged for 15 min through a 0.22-µm-pore-size Spin X cellulose acetate spin filter (Corning Costar, Merck, Darmstadt, Germany).

LC-MS/MS analysis, database search, and data analysis: LC-MS/MS analysis of trypsin-shaved surface peptides was performed on an Ultimate 3000 RSLC nanoLC instrument coupled to a QExactive HF mass spectrometer (Thermo-Fisher Scientific, Waltham, MA, USA) as described previously [15]. MS/MS data were searched against the *A*. *fumigatus* Af293 database of the AspGD (http://www.aspergillusgenome.org/download/sequence/A_fumigatus_Af293/archive/A_fumigatus_Af293_version_s03-m05-r16_orf_trans_all.fasta.gz, accessed on 3 November 2019) using Proteome discoverer 2.2 and the algorithms of Mascot 2.4.1, Sequest HT, and MS Amanda 2.0. Two missed cleavages were allowed for tryptic peptides, the precursor mass tolerance was set to 10 ppm, and the fragment mass tolerance was set to 0.02 Da. The dynamic modification was oxidation of Met. Only rank-1 peptides were counted exclusively for top scored proteins and a strict target false-discovery rate (FDR) of <1% (calculated against a reverse decoy database) on both peptide and protein level was required. Search engine score thresholds were >30 for Mascot, >300 for MS Amanda, and >4 for Sequest. The mass spectrometry proteomics data have been deposited to the ProteomieXchange Consortium via the PRIDE [19] partner repository with the dataset identifier (PXD020136).

Stimulation of human dendritic cells: Monocytes isolated from the peripheral blood mononuclear cells (PBMC) of healthy donors by using CD14 MicroBeads (Miltenyi Biotec, Paris, France) were cultured in RPMI 1640 medium with 10% fetal calf serum, GM-CSF (1000 IU/million cells) and IL-4 (500 IU/million cells) (Miltenyi Biotec, Paris, France) for six days. Differentiation of DCs was confirmed by the expression of CD1a and the lack of surface expression of CD14 and CD16 [20]. DCs were stimulated with conidia or cell wall polysaccharides as described earlier [1,9,21]. β-(1,3)-Glucan and α-(1,3)-glucan were used at equal absolute concentrations (1 μg); they varied in their chain size, thus ruling out the possibility of using them at equimolar concentrations. The yield of α-(1,3)-glucan and β-(1,3)-glucan from 1 × 10^6^ conidia was approximately 1.1 and 3 μg, respectively. DCs (5 × 10^5^/well) were incubated with 5 × 10^5^ conidia or polysaccharides (1 μg) per well for 48 h at 37 °C in a CO_2_ incubator. The cell-free culture supernatants were collected and analyzed for cytokines. The DC-phenotype was determined by flow cytometry analysis of the surface markers by using fluorochrome-conjugated monoclonal antibodies to CD80 (BD Biosciences) and CD40 (Beckman Coulter, Villepinte, France). The values were presented as intensity of expression of the molecules depicted as median fluorescence intensity (MFI). In another set of experiment, Dectin-1 on DCs was blocked with an anti-Dectin-1 monoclonal antibody (GE2; 8 μg/0.5 × 10^6^ DCs) [22] for 30 min, followed by stimulation with ∆*rodA* conidia. Cytokines (IL-10 and IL-8) in the culture supernatants were quantified using ELISA (Ready-SET-Go, eBioscience, Paris, France).

Nuclear Magnetic Resonance (NMR) analyses: NMR experiments were performed on a Direct Drive 600 MHz spectrometer (Agilent Technologies, Santa Clara, CA, USA) equipped with a cryogenic probe. ^1^H-^15^N HSQC (heteronuclear single quantum coherence) experiments were recorded on samples of ^15^N labeled recombinant RodAp. The spectrum of the folded protein in 20 mM sodium acetate pH 4.3 10% D_2_O was reported previously [7]. To obtain the spectrum of RodAp without disulfide bonds, the buffer was supplemented with 5 mM TCEP (Tris (2-carboxyethyl)phosphine) to reduce the cysteine residues. ^15^N labeled recombinant RodAp was expressed and purified as described [9]. Spectra recorded with VnmrJ 4.3 (Agilent), processed with NMRPipe, and analyzed with CCPNMR analysis.

Statistical analysis: Performed using GraphPad-prism (Version 8). The significance of differences between series of results was assessed using the two-tailed Mann-Whitney *U*-test or one-way ANOVA, and multiple comparison between sets of results was assessed using Dunnett’s post-test.

## 3. Results

### 3.1. Disulfide Bonds are Necessary for the Structure of the Spontaneously Self-Assembling RodAp

Earlier, we showed that (i) the structure of RodAp monomers that are competent to assemble into rodlets is organized around four disulfide bonds formed by the eight conserved cysteine residues in the RodAp sequence [7] and that (ii) mutation of either one or four cysteine residues resulting in the disruption of one or all the four disulfide linkages generated mutant *A*. *fumigatus* conidia devoid of surface rodlets [9]. To understand the effect of the absence of disulfide bonds on the structure of RodAp, using NMR we analyzed the effect of the reduction of the disulfide bonds on the structure of the recombinant protein. We used ^15^N labeled RodAp to record ^1^H-^15^N correlation spectra (HSQC) that produce a signal for each residue backbone amide group. The chemical shifts (frequencies) of the amide groups are exquisitely sensitive to the chemical environment and can be considered as the fingerprint of a structure (Figure 1). Reduction of the cysteines with the reducing agent TCEP abrogates the disulfide bonds and has a drastic effect on the HSQC spectrum of recombinant RodAp. Indeed, the spectrum of the reduced protein displays a very low dispersion of signals in the proton dimension and a reduced number of signals indicative of a disordered protein. This result shows that without disulfide bonds, RodAp is unfolded.

### 3.2. Point-Mutations of the Conserved Cysteine Residues of RODA Resulted in the Lack of the Rodlet Layer on the Conidial Surface

Knowing that no rodlets are present at the surface of *A*. *fumigatus* mutants with one or four disrupted disulfide bridges [9], we subjected the conidial cytoplasmic fraction to Western blot using anti-RodA antibodies. The conidial cytoplasmic contents of the single (C127S), as well as the quadruple (C64S/C65S/C133S/C134S) ccrp**-**mutant conidia failed to show any positive band for RodAp, unlike the cytoplasmic content of the parental strain (Figure 2A).

### 3.3. Absence of Rodlets Exposes α-(1,3)-Glucan on the Conidial Surface

We then sought to study the impact of the absence of the rodlet layer on the RodA-mutant conidial surfaces. First, we looked at the conidial cell wall permeability. When ccrp-mutant conidia were labeled with Calcofluor-White (CFW) or fluorescein isothiocyanate (FITC), both the single and quadruple ccrp-mutant conidia showed a similar labeling pattern to the parental strain conidia; only surface labeling of the conidia was observed (Figure 2B; FITC labeling has been reported). Likewise, the ∆*rodA* mutant conidia only displayed surface labeling with CFW or FITC (Figure 2B, shows FITC labeling), suggesting that the absence of the surface rodlet layer either due to cysteine point mutation or gene-deletion does not alter the permeability of mutant conidia.

Further, we analyzed the cell wall components exposed on the conidial surfaces. In the parental strain conidia, the melanin layer lies immediately beneath the rodlet layer [23]. Immunolabeling showed punctate exposure of melanin on the parental strain conidia (Figure 3A), whereas in ccrp-mutants and ∆*rodA* conidia, the melanin layer was exposed on the conidial surface (Figure 3A). Quantification of the conidial melanin content indicated that there was 4.1 ± 0.2, 1.9 ± 0.3 and 2.1 ± 0.2-fold increase in the ∆*rodA*, single and quadruple ccrp-mutants, respectively, compared to the parental strain conidia. We then investigated the exposure of other cell wall polysaccharides on the conidial surfaces. ConA-FITC labeling was negative or weakly positive for the parental strain, ccrp-mutants and ∆*rodA* conidia. In addition, there was weakly positive WGA-FITC labeling of ccrp-mutant and ∆*rodA* conidia, but not of parental strain conidia. Both single and quadruple ccrp-mutant conidia were strongly positive for α-(1,3)-glucan and only weakly positive for β-(1,3)-glucan, ∆*rodA* conidia were weakly positive for α-(1,3)-glucan and strongly positive for β-(1,3)-glucan, whereas parental strain conidia were negative for both polysaccharides (Figure 3B,C). Differential conidial labeling for β-(1,3)-glucan and α-(1,3)-glucan was further confirmed by flow cytometric analyses (Figure 3D).

### 3.4. The Proteins Exposed on the Ccrp-Mutant Conidial Surface are More Similar to Those of the Parental Strain

Table A1 represents the scores of the peptide-spectrum matches (PSM; the total number of identified peptide spectra matched for a protein) for proteins cleaved off from the conidial surfaces after subjecting to a brief trypsin digestion and proteomic analysis of these extracted proteins. The PSM score for RodA in the parental strain was 160 ± 111, while the protein was absent in the ∆*rodA* mutant conidial extract; both single and quadruple ccrp-mutant showed intermediary values (36 ± 13 and 31 ± 7, respectively) suggesting that the point mutations significantly affect the secretion of RodAp to the conidial surface. We compared the proteins qualitatively; the parental strain and the ∆*rodA* conidia had unique proteins (Figure 4), whereas the single and quadruple ccrp-mutants had only two and one unique proteins, respectively. In total, 34, 38, 22, and 13 proteins were identified in the surface protein extracts from the parental strain, ∆*rodA*, single and quadruple ccrp-mutant conidia, respectively. The ∆*rodA* conidial protein profile showed only 48% similarity with the profile of the parental strain. However, among the common proteins identified, the single ccrp-mutant showed 64% and 28% similarity with the parental strain and ∆*rodA*, respectively, while it was 77% and 15% for the quadruple ccrp-mutant with that of the parental strain and ∆*rodA* extracts, respectively. These results suggest that the conidial surface exposed protein-composition of the ccrp-mutant conidia more closely resemble the parental strain than the ∆*rodA* conidia.

### 3.5. Ccrp-Mutant Conidia Are Less Stimulatory than ∆rodA Conidia

Differences in the surface exposed polysaccharides of ∆*rodA* conidia compared to ccrp-mutant conidia raised the possibilities that these conidia might show differences in their ability to stimulate immune cells. We thus investigated the ability of these conidia to induce maturation of human DCs by analyzing the expression of CD40 and CD80, the co-stimulatory molecules, which are enhanced upon receipt of activation signals by DCs. We found that both ∆*rodA* and ccrp-mutant conidia have equivalent abilities to induce CD40 and CD80 (Figure 5A). As expected, the dormant conidia of the parental strain did not induce maturation markers on DCs [1]. However, analyses of IL-10 and IL-8 cytokines revealed that ∆*rodA* conidia were more stimulatory than ccrp-mutant conidia (Figure 5B). Indeed, while ∆*rodA* conidia induced up to 78-fold and 465-fold increases in IL-10 and IL-8 compared to the untreated control DCs, ccrp-mutant conidia (single/quadruple mutants) induced only 26-fold and 28-fold increases in IL-10 and IL-8, respectively. Single and the quadruple ccrp-mutant conidia displayed a similar induction of IL-10 and IL-8 (Figure 5B).

Since the conidial surface of ccrp-mutants strongly exposed α-(1,3)-glucan and that of ∆*rodA* conidia was strongly positive for β-(1,3)-glucan, we explored whether differences in the ability of these mutant conidia to induce DC cytokines could be attributed to these surface exposed polysaccharides. Therefore, we stimulated DCs with α-(1,3)-glucan or β-(1,3)-glucan isolated from the conidial cell wall of the parental strain [24], and analyzed for the expression of co-stimulatory molecules on DCs and the secretion of IL-10 and IL-8. Though α-(1,3)-glucan and β-(1,3)-glucan stimulated the expression of CD40 and CD80 to a similar extent (Figure 5C), β-(1,3)-glucan was a stronger inducer of DC cytokine IL-10 compared to α-(1,3)-glucan (Figure 5D). However, both glucans induced similar amounts of IL-8. Together, these data imply that surface exposure of polysaccharides contributes at least in part for the differential DC-stimulatory ability of ∆*rodA* and ccrp-mutant conidia. To test this proposition, we blocked Dectin-1, a C-type lectin receptor implicated in the recognition of β-(1,3)-glucan [4,22] and then stimulated the DCs with ∆*rodA* conidia. However, analyses of DC cytokines revealed that Dectin-1 blockade had no repercussion on the secretion of both IL-10 and IL-8 (Figure 5E), implying that β-(1,3)-glucan is responsible only to a minor extent for the stimulation of DCs by ∆*rodA* conidia or it is recognized by Dectin-1 independent pathway [25].

### 3.6. Chemical Removal of the Rodlet Layer Results in the Exposure of α-(1,3)-Glucan on the Conidial Surface

We then chemically removed the surface rodlet layer from the parental strain conidia using HF and immunolabeled the resultant conidia to expose the melanin pigment, β-(1,3)-glucan and α-(1,3)-glucan. The labeling pattern of surface exposed α-(1,3)-glucan on HF-treated conidia was similar to those of the ccrp-mutants (Figure 6A), but there was no exposure of β-(1,3)-glucan on these HF-treated conidial surfaces. These observations were further confirmed by flow-cytometry, wherein HF-treated conidia were positive for α-(1,3)-glucan labeling (Figure 6B). Although weak, the HF-treated conidia showed positive labeling for melanin, possibly due to disintegration of the melanin layer upon HF-treatment. Indeed, HF-treatment beyond 36 h abrogated melanin labeling.

### 3.7. Cell Wall Components Exposed on the Ccrp-Mutant and Swollen Conidia Are Similar

Swollen conidia (obtained after 2 h/4 h incubation of parental dormant conidia in the culture medium) were labeled to localize melanin, β-(1,3)-glucan and α-(1,3)-glucan with MelLec, Dectin-1, and anti-α-(1,3)-glucan-serum, respectively. We observed a gradual decrease in the intensity of the immunolabeling for melanin in accordance with its disintegration during the conidial swelling process (Figure 7 compared to Figure 3). Concomitantly, the fluorescent intensity for α-(1,3)-glucan increased, in agreement with the earlier study showing the presence of α-(1,3)-glucan on the swollen conidial surface [26]. In the case of β-(1,3)-glucan, though there was an intensification of the labeling with the swelling process (from 2 to 4 h), the labeling remained feeble. Together, surface labeling of swollen conidia was similar to that of ccrp-mutants.

## 4. Discussions

In this study, we compared two different genetic strategies of obtaining *A*. *fumigatus* conidia devoid of the surface rodlet layer, either deleting the gene expressing RodAp involved in rodlet formation or by engineered point-mutation of the RodA-conserved cysteine residues. Both strategies resulted in mutant conidia devoid of rodlets, but RodAp was still present on the ccrp-mutant conidial surface. However, their surface properties and immunostimulatory potentials were significantly different. Possibly as a compensatory mechanism due to the loss of the surface rodlet layer, the mutant conidia had increased melanin content compared to the parental strain conidia irrespective of the RodAp deletion strategy used. Nevertheless, this increase was significantly higher in the ∆*rodA* conidia than in the single or quadruple-ccrp mutants. The ∆*rodA* conidia showed higher β-(1,3)-glucan and lower α-(1,3)-glucan exposure on their conidial surface; whereas, this phenotype regarding β-(1,3)-glucan and α-(1,3)-glucan exposure on the conidial surface were flipped in the ccrp-mutants compared to ∆*rodA*. As observed before, the parental strain conidia did not expose either glucans, as these were masked by the rodlet layer [1]. The inactivated conidia of the parent strain failed to activate DCs, in accordance with our previous observation [1]. Both ∆*rodA* and ccrp*-*mutant conidia, however, promoted the maturation of DCs and stimulated the secretion of cytokines from the DCs. Though these mutant conidia had equivalent capacities to induce DC maturation markers, they displayed significant differences in their capacity to stimulate DC activation.

Probing for RodAp in the cytoplasm of the ccrp-mutants using anti-RodAp antibodies was negative on Western blots, whereas proteomic analyses of the conidial surface tryptic digests showed the presence of RodAp on the ccrp-mutants, albeit with significantly lower PSM scores relative to the parental strain tryptic digest. We did not detect any patch of rodlets on the ccrp-mutants upon subjecting spores to atomic force microscopy [7]. These observations suggest that the RodA-conserved cysteine residues are essential for the proper folding and stability of RodAp, which in turn may be necessary for the transport of RodAp to the conidial surfaces and organization into a structured rodlet layer.

The deletion strategy to characterize the function of a gene encoding the fungal cell wall protein is frequently associated with compensatory mechanisms, obscuring the exact function of the deleted gene. For example, in *A*. *fumigatus*, the deletion of AGS (encoding for the α-(1,3)-glucan synthases) genes led to the deposition of an amorphous glycoprotein matrix above the rodlet layer of the mutant conidia [27]. The functional loss of FKS1 (encoding for the β-(1,3)-glucan synthase) and compensatory increases of chitin and galactomannan in the cell wall, yet, shedding a massive amount of galactomannan from the cell wall [28]. The deletion of mannosyltransferases reduces mannan content in the conidial cell wall, affecting the cell wall organization, permeability and conidial viability [29]. Such compensations are also seen for the CcpAp of *A*. *fumigatus* [15] as well as other pathogenic fungi, such as *Candida albicans* and *Cryptococcus neoformans*. The deletion of KRE5 (required for β-(1,6)-glucan synthesis) in *C*. *albicans* leads to increased level of chitin and decreased levels of mannoprotein in the cell wall [30]. In *C*. *neoformans*, the deletion of KRE genes does not affect the amount of chitin, but alters its localization in the cell wall [31]. In agreement with the above observations, the removal of the conidial surface rodlet layer, either through RODA deletion or through point mutation of the RodAp-conserved cysteine residues, also resulted in a compensatory increase in the conidial melanin content. Noteworthy, the exposure of conidial cell wall polysaccharides that are otherwise masked by the rodlet layer was different with the mutation strategy used.

The consequences of modifying a fungal cell wall gene on the host-fungal interaction need to be interpreted carefully. The removal of the conidial surface rodlet layer by gene deletion or point mutation of the conserved cysteine residues resulted in differential exposure of the conidial cell wall polysaccharides. Accordingly, these mutants showed different immunostimulatory properties; ∆*rodA* conidia promoted significantly higher induction of cytokine secretion by DCs, compared to DC-stimulated with single/quadruple ccrp-mutants. When we characterized the polysaccharides exposed on the surface during conidial swelling, a process prior to conidial germination, as reported earlier, even though there was a stage-specific exposure of β-(1,3)-glucan [4], the dormant conidia after chemical removal of its surface rodlet layer and the swollen conidia showed higher surface exposure of α-(1,3)-glucan. The proteomic analysis of the conidial surface-exposed proteins indicated that the protein-profiles of the ccrp-mutants are more similar to the conidial proteomic profile of the parental strain. On the contrary, the ∆*rodA* conidial surface proteome showed a distinct set of proteins compared to that of the parental strain. Altogether, the analyses of surface exposed polysaccharides and proteins indicate generation of a physiologically novel conidial phenotype upon ccrp-mutations of RODA. Therefore, generation of a rodletless mutant by point mutations of the RodA-conserved cysteine residues could be a better strategy to obtain a mutant without major compensatory modifications in the cell wall.

Though β-(1,3)-glucan strongly induced DC cytokines, ∆*rodA* conidia were superior to β-(1,3)-glucan alone in their ability to induce cytokines. One of the reasons could be that β-(1,3)-glucan on the surface of conidia is presented in a different three-dimensional organization with an enhanced DC-stimulatory capacity compared to isolated β-(1,3)-glucan. Additional polysaccharides and/or proteins on the surface of ∆*rodA* conidia or co-stimulation through both Dectin-1 and other pattern recognition receptors [32] might also contribute for the enhanced stimulatory ability of ∆*rodA* mutant conidia. However, blocking of Dectin-1 on DCs did not alter the activation or cytokine secretion by DCs upon interaction with ∆*rodA* conidia. Nevertheless, our data clearly demonstrate that compared to α-(1,3)-glucan, β-(1,3)-glucan strongly stimulates DC cytokines IL-10 and IL-8. These results are in line with our previous data demonstrating that β-(1,3)-glucan induces high levels of IL-12 in DCs, a key cytokine for the polarization of Th1 responses. Accordingly, although both β-(1,3)-glucan and α-(1,3)-glucan induced comparative levels of PD-L1 on DCs, β-(1,3)-glucan strongly promoted DC-mediated Th1 and regulatory T cell (Treg) responses, whereas α-(1,3)-glucan promoted only Treg responses [21].

Until now the studies have focused on the stage-specific exposure of β-(1,3)-glucan on the *A*. *fumigatus* conidial surfaces during the germination. Whereas, our study indicates that compared to β-(1,3)-glucan, α-(1,3)-glucan is proportionately more exposed on the conidial surfaces during germination, suggesting a need to study the consequences of state-specific exposure of α-(1,3)-glucan during conidial germination during immune cells and conidial interactions. Indeed, a recent study of the mycelial cell wall by solid-state NMR indicated that α-(1,3)-glucan is the superficial polysaccharide present in the cell wall [33]. α-(1,3)-Glucan is recognized by TLR2 [21] and that conidial internalization is associated with TLR2 recruitment in the phagosomes [34], further indicate the importance of the α-(1,3)-glucan exposure.

## 5. Conclusions

It is a common practice to use gene-deletion mutant as a control while establishing functional role associated with a fungal gene. However, gene-deletion is often associated with compensatory modifications in the mutant fungus, which obscures function of the deleted gene and particularly impacts host-fungal interaction studies. Our study suggests that point mutations of the conserved or essential amino acids of the fungal cell wall associated genes could be an alternative strategy to study the functions associated with the respective gene-expression products, as they may not lead to major compensatory modifications in the cell wall compared to the gene-deletion strategy that leads to a drastic alteration in the cell wall organization. Moreover, the stage-specific exposure of β-(1,3)-glucan during conidial germination has been studied in the context of immunomodulation by *A*. *fumigatus*. Our study of removal of rodlet-layer genetically (by point-mutation of the cysteine residues of RodAp), chemically (by HF-treatment) or biologically (swollen conidia) suggests that α-(1,3)-glucan is located beneath the rodlet layer, which potentially interacts with the immune system.

## Figures and Tables

**Figure 1 jof-06-00151-f001:**
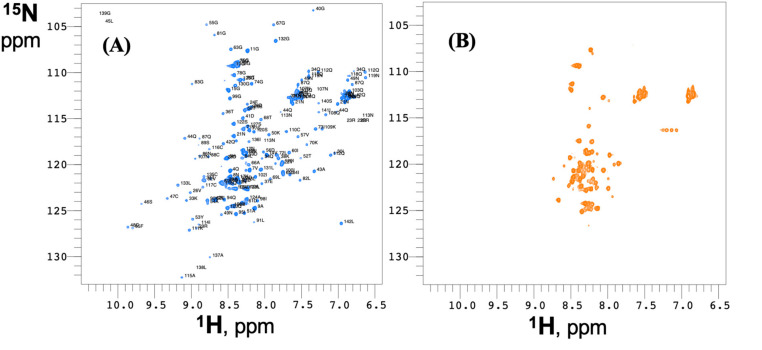
^1^H-^15^N correlation spectra (HSQC) spectra of (**A**) folded RodAp and (**B**) reduced RodAp. In (**A**), the residue and number of each backbone amide correlation is shown. Spectra were recorded at 25 °C in 20 mM sodium acetate buffer (pH 4.3) with 5% D_2_O. To reduce the cysteine residues of the protein, 5 mM TCEP was added to the buffer.

**Figure 2 jof-06-00151-f002:**
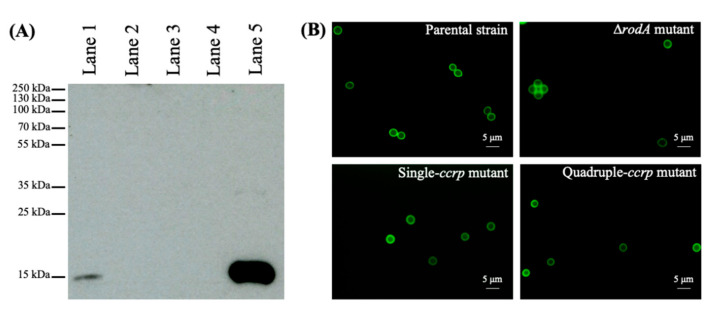
(**A**) Western blot analysis of the cytoplasm contents of conidia for RodAp probed with polyclonal anti-RodAp antibodies; recombinant RodAp was used as the control (lane 5); lanes 1–4 represent the cytoplasmic contents from the parental strain, ∆*rodA*, single and quadruple ccrp-mutants, respectively. Only the conidial cytoplasm of the parental strain (Lane 1) showed a positive band for RodAp. (**B**) Conidial labeling with fluorescein isothiocyanate; both ∆*rodA* and ccrp-mutant conidia showed only surface labeling similar to parental strain conidia, suggesting that the cell wall permeability of the rodletless mutants (by gene deletion or RodAp-point mutations) were unaltered compared to parental strain conidia.

**Figure 3 jof-06-00151-f003:**
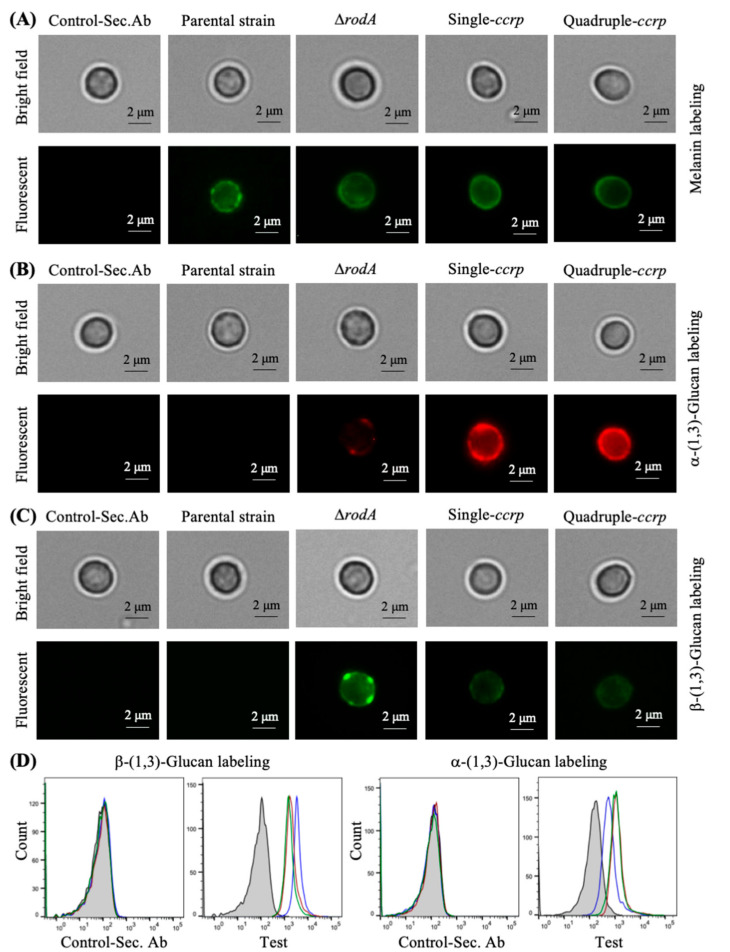
Immunolabeling of parental strain, ∆*rodA* and ccrp-mutant dormant conidia with MelLec, Dectin-1 (both human Fc-conjugated), and polyclonal anti-α-(1,3)-glucan-serum (melanin (**A**), α-(1,3)-glucan (**B**) and β-(1,3)-glucan (**C**), respectively); the controls were conidia treated with secondary antibodies (Fc-specific anti-human-IgG-FITC for MelLec and Dectin-1, and anti-mouse-IgG-TRITC for α-(1,3)-glucan). Parental strain conidia showed punctate labeling for MelLec and were negative for β-(1,3)-/α-(1,3)-glucan, ∆*rodA* conidia showed uniform labeling for MelLec, intense labeling for β-(1,3)-glucan and weak labeling for α-(1,3)-glucan; while both single and quadruple *ccrp*-mutants showed strong labeling for α-(1,3)-glucan but weak β-(1,3)-glucan labeling. (**D**) Flow cytometry upon labeling conidia for β-(1,3)-glucan and α-(1,3)-glucan (two biological replicates; in gray-filled—parental strain, in blue line—∆*rodA*, single-ccrp*—*in red line and quadruple-ccrp—in green line; controls were conidia treated with only FITC-conjugated secondary antibodies) (in the spectra shown, median fluorescence intensities were, for β-(1,3)-glucan—58, 3314, 1685 and 1364, respectively, for parental strain, ∆*rodA*, single and quadruple-ccrp mutants, and for α-(1,3)-glucan—135, 544, 1069, and 1017, respectively, for parental strain, ∆*rodA*, single and quadruple-ccrp mutants).

**Figure 4 jof-06-00151-f004:**
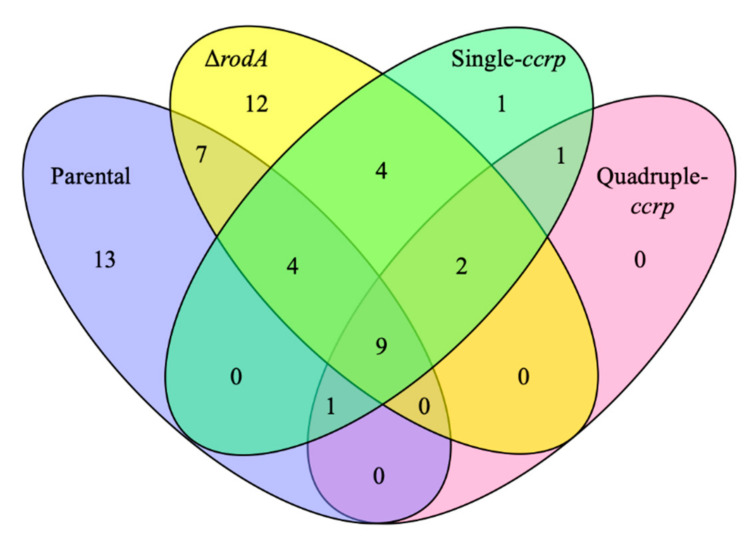
Four-ways Venn-diagram showing the number of proteins identified in the extract after subjecting the parental strain, ∆*rodA* and ccrp-mutant dormant conidia to a brief trypsin digestion followed by proteomic analysis of the extracted proteins using LC-MS/MS. There were 34, 38, 22, and 13 proteins identified in the parental strain, ∆*rodA*, single and quadruple ccrp-mutant conidial extracts, respectively. Of the total number of proteins extracted, the single ccrp-mutant showed 64% and 28% of similarity and the quadruple ccrp-mutant showed 77% and 15% similarity with the parental strain and ∆*rodA*, respectively.

**Figure 5 jof-06-00151-f005:**
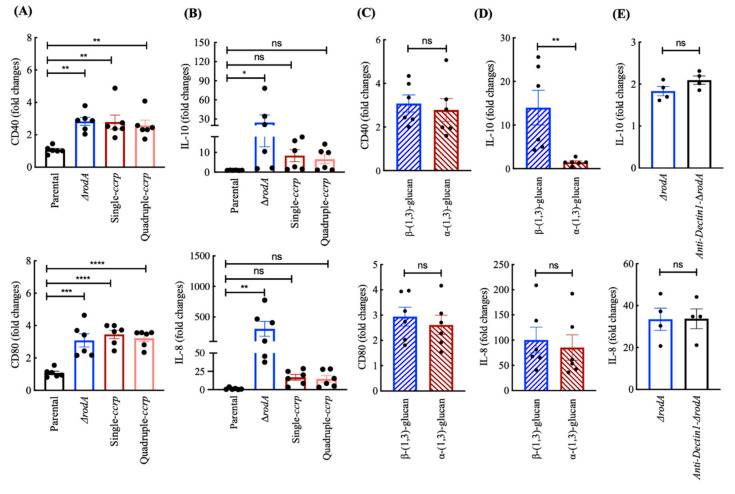
Conidial immunostimulatory capacities: (**A**) monocyte-derived DCs were cultured either in medium alone or with dormant conidia, ∆*rodA* or ccrp-mutant conidia for 48 h. The expression of CD80 and CD40 on DCs was analyzed by flow cytometry. Data (mean ± SEM, *n* = 6 donors) are presented as fold changes in the median fluorescence intensity compared to untreated DCs. (**B**) Fold changes (mean ± SEM, *n* = 6 donors) in the IL-10 and IL-8 amounts in the cell-free culture supernatants compared to untreated control DCs. Statistical significance as determined by one-way ANOVA; * *p* < 0.05, ** *p* < 0.01, *** *p* < 0.001 **** *p* < 0.0001 and ns, not significant. (**C**,**D**) Comparison of the DC-stimulatory functions of β-(1,3)-glucan and α-(1,3)-glucan. DCs were stimulated with isolated α-(1,3)-glucan or β-(1,3)-glucan for 48 h and analyzed for the expression of CD80 and CD40 (**C**) and the secretion of IL-10 and IL-8 (**D**). Data (mean ± SEM, *n* = 6 donors) are presented as fold changes in the cytokines compared to untreated control DCs. Statistical significance as determined by two-way Mann-Whitney *U*-test; ** *p* < 0.01 and ns, not significant. (**E**) DCs were treated with an anti-Dectin-1 MAb followed by stimulation with ∆*rodA* conidia. Cytokines (IL-10 and IL-8) in the cell-free culture supernatants were quantified and data (mean ± SEM, *n* = 2 donors in duplicates) are presented as fold changes in cytokines compared to untreated control DCs. Statistical significance as determined by two-way Mann-Whitney *U*-test; ns, not significant.

**Figure 6 jof-06-00151-f006:**
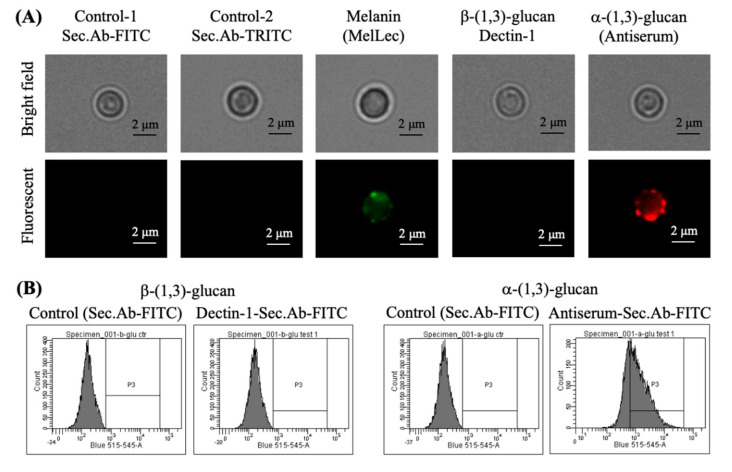
(**A**) Immunolabeling of parental strain conidia after the chemical removal of the rodlet layer using hydrofluoric acid (hydrofluoric acid (HF)-treated conidia); labeling was performed with MelLec, Dectin-1 (both human-Fc conjugated), and polyclonal anti-α-(1,3)-glucan-serum for melanin, β-(1,3)-glucan and α-(1,3)-glucan, respectively; controls were the HF-treated conidia incubated with secondary antibodies [anti-human-IgG-FITC for MelLec and Dectin-1 (in green), and anti-mouse-IgG-TRITC for α-(1,3)-glucan (in red)]. HF-treated conidia showed strong labeling for α-(1,3)-glucan. (**B**) Flow-cytometric analyses of HF-treated conidia labeled for β-(1,3)-glucan (with Dectin-1) and α-(1,3)-glucan (using polyclonal antiserum); controls were the HF-treated conidia incubated with secondary antibodies (Fc-specific anti-human-IgG-FITC for Dectin-1 and anti-mouse-IgG-FITC for α-(1,3)-glucan) (in the spectra shown, median fluorescence intensities for control and Dectin-1 or antiserum labeled conidia were, for β-(1,3)-glucan—148 and 147, respectively, and for α-(1,3)-glucan—145 and 899, respectively).

**Figure 7 jof-06-00151-f007:**
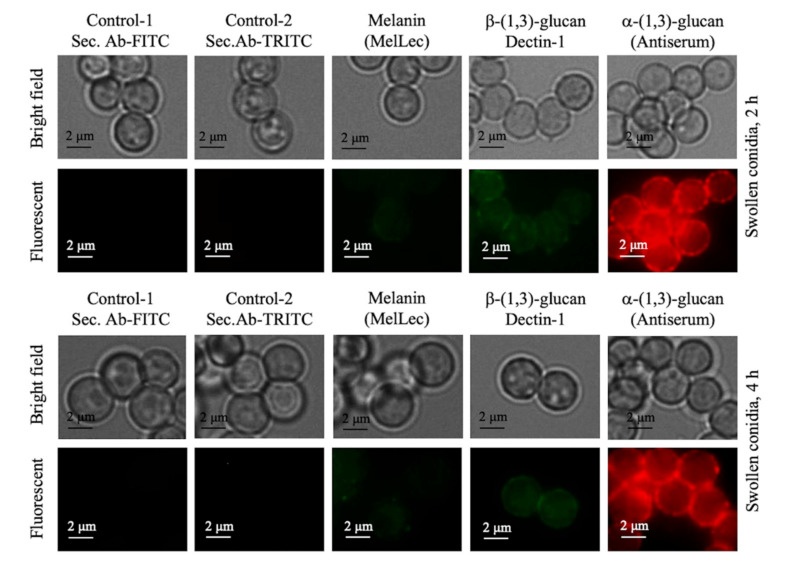
Swollen conidia (after 2 or 4 h of dormant conidial incubation in liquid culture medium) immunolabeled with MelLec, Dectin-1, and polyclonal anti-α-(1,3)-glucan-serum for melanin, β-(1,3)-glucan and α-(1,3)-glucan detection, respectively; the controls were swollen conidia treated with secondary antibodies [anti-human-IgG-FITC for MelLec/Dectin-1 (green), and anti-mouse-IgG-TRITC for α-(1,3)-glucan (in red)]. During the course of germination (from 2 to 4 h), there was a gradual decrease in the MelLec labeling, due to the disintegration of melanin pigments, while there was intensification of the labeling for α-(1,3)-glucan; although there was also intensification of the Dectin-1 labeling for β-(1,3)-glucan, the labeling was weak compared to that of α-(1,3)-glucan labeling.

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
