# Peer review of "The Role of RodA-Conserved Cysteine Residues in the Aspergillus fumigatus Conidial Surface Organization"

_jof, 2020, doi:10.3390/jof6030151_

Round 1
Reviewer 1 Report
The study by Valsecchi, et al. is a small follow up to the initial study of Aspergillus conidia that lack the RodA rodlet protein. In this study, point mutations disrupting the folding of RodA are used instead of deletion of the gene encoding RodA. As might be expected, this leads to phenotypes intermediate between the wild-type and the complete loss of RodA.
Issues to addressed
The authors argue the intermeidate phenotypes are an advantage since it avoids some of the compensatory changes in cell wall organization caused by the complete loss of RodA. However, the “milder” phenotypes may cause certain functions to be missed as well as the creation of physiologically-novel phenotypes (e.g., high exposure of alpha-1,3-glucan) that may confuse interpretation of the normal roles of a protein. These caveats should be addressed in the discussion.
Figure 5. The authors would like to attribute the greater stimulation of DCs by mutant conidia to the greater exposure of polysaccharides. To test this, they add soluble glucans and see stimulation primarily by beta-1,3-glucan. However, these are presented in an unnatural context (which the authors acknowledge in lines 380-384). These amount to correlations, not evidence that the exposed beta-glucans stimulated the DCs. The authors should use blocking of beta-glucan receptors or use of beta-glucan competitors (e.g., laminarin) to interfere specifically with sensing of beta-glucans to show that the mutant conidia are stimulating via the exposed-beta-glucans.
Significant parts of the discussion only reiterate already described results (lines 317-332, 365-374) instead of putting them into a broader context.
Other corrections/suggestions
line 83: Provide brief description of the relevant aspects of the melanin extraction procedure instead of just providing the reference.
line 94 Provide the specific antibody used to detect alpha-1,3,-glucan
line 133 Provide brief description of how DCs were derived from PBMCs as CD40 and CD80 can also be present on macrophages
line 319 “point-mutating” is laboratory slang. Replace with “engineering point mutations”
line 320 while conidia lack rodlets, it’s worth noting to the reader that RodA protein is still present on the surface of ccrp mutant conidia
line 326 As used, “reversed” implies reversion to wild-type levels and polysaccharides do not return to wild-type (and in-fact the alpha-1,3-glucan exposure was much higher than wild-type or the rodA). Instead, the phenotypes regarding alpha- and beta-1,3-glucans were flipped compared to the rodA deletion.
lines 337-342 No data is suggestive that vesicular-transport of RodA requires proper folding, especially since no internal RodA was found. The data is more consistent that RodA folding is necessary for rodlet formation.
line 343 replace “fungal cell wall gene” with “gene encoding the fungal cell wall protein”
lines 385-391 tangential discussion
line 416 add “potentially” to alpha-1,3-glucan interaction with the immune system. No functional/consequential evidence of such an interaction has been well-documented.
Author Response
Reviewer 1
Issues to addressed
The authors argue the intermediate phenotypes are an advantage since it avoids some of the compensatory changes in cell wall organization caused by the complete loss of RodA. However, the “milder” phenotypes may cause certain functions to be missed as well as the creation of physiologically-novel phenotypes (e.g., high exposure of alpha-1,3-glucan) that may confuse interpretation of the normal roles of a protein. These caveats should be addressed in the discussion.
Response: We have addressed the caveats identified by the reviewer.
Figure 5. The authors would like to attribute the greater stimulation of DCs by mutant conidia to the greater exposure of polysaccharides. To test this, they add soluble glucans and see stimulation primarily by beta-1,3-glucan. However, these are presented in an unnatural context (which the authors acknowledge in lines 380-384). These amount to correlations, not evidence that the exposed beta-glucans stimulated the DCs. The authors should use blocking of beta-glucan receptors or use of beta-glucan competitors (e.g., laminarin) to interfere specifically with sensing of beta-glucans to show that the mutant conidia are stimulating via the exposed-beta-glucans.
Response: As this reviewer correctly noted, we did not claim that differential surface exposure of beta-1,3-glucan was responsible for the greater stimulation of DCs by RodA mutant conidia rather it was one of the factors. In line with suggestion of the reviewer, we blocked Dectin-1 on DCs with monoclonal antibody followed by stimulation of the cells with rodA mutant conidia. The data is provided in the Figure 5, and has been discussed in the text.
Significant parts of the discussion only reiterate already described results (lines 317-332, 365-374) instead of putting them into a broader context.
Response: In line with suggestions of the reviewer, we have now edited the discussion.
Other corrections/suggestions
line 83: Provide brief description of the relevant aspects of the melanin extraction procedure instead of just providing the reference.
Response: A brief explanation is provided in the methods section of the revised manuscript.
line 94 Provide the specific antibody used to detect alpha-1,3,-glucan
Response: It was provided before (Ref.14), but the sentence has been revised for clarity.
line 133 Provide brief description of how DCs were derived from PBMCs as CD40 and CD80 can also be present on macrophages
Response: The description is now included the revised manuscript.
line 319 “point-mutating” is laboratory slang. Replace with “engineering point mutations”
Response: Replaced.
line 320 while conidia lack rodlets, it’s worth noting to the reader that RodA protein is still present on the surface of ccrp mutant conidia
Response: Modified the text indicating that ccrp-mutants still show the presence of RodAp in their conidial surface-extractable proteins.
line 326 As used, “reversed” implies reversion to wild-type levels and polysaccharides do not return to wild-type (and in-fact the alpha-1,3-glucan exposure was much higher than wild-type or the rodA). Instead, the phenotypes regarding alpha- and beta-1,3-glucans were flipped compared to the rodA deletion.
Response: In line with previous comment on the reiteration of already described results, we have now edited the discussion and hence above-mentioned sentence has been removed.
lines 337-342 No data is suggestive that vesicular-transport of RodA requires proper folding, especially since no internal RodA was found. The data is more consistent that RodA folding is necessary for rodlet formation.
Response: We deleted this ambiguous/hypothetical explanation.
line 343 replace “fungal cell wall gene” with “gene encoding the fungal cell wall protein”
Response: Modified as suggested.
lines 385-391 tangential discussion
Response: We have now edited the discussion.
line 416 add “potentially” to alpha-1,3-glucan interaction with the immune system. No functional/consequential evidence of such an interaction has been well-documented.
Response: Added the word ‘potentially’.
Reviewer 2 Report
In this manuscript by Valsecchi et al they examine the immunostimulatory capacity of the cccp mutants in the RodAp versus the complete rodA-null mutant strain of Aspergillus fumigatus. The authors find that the cccp mutants are less immune stimulatory in terms of cytokines (IL-10 and IL-8), but induce similar levels of DC co-stimulatory molecules. The authors go on to try to demonstrate that this is due to the difference in alpha-1,3-glucan and beta-1,3-glucan exposure in the cells wall. This seems plausible but there are a couple controls that would strengthen their arguments and conclusions partially in regards to the glucan stimulations of the DCs. The biggest concern I have is the authors keep saying the cccp mutant does not have RodAp on the cell surface, but there proteome data argues against that.
1) Section 3.2: I would argue based on the data from the proteomic analysis (Section 3.4) that the RodAp protein is in the cell wall (on the cell surface) somehow since you detect the presence of RodAp by LC-MS/MS, albeit about 5-fold reduced. I think this fact is very important for the interpretation of all the results especially the HF striping of the rodlets later in the manuscript, when those spore look more like the cccp mutant than the rodA-null mutant.
2) Figure Legend 3: It appears the panel labels or the legend label are mixed up for which set of panels is b1,3-glucan vs. a1,3-glucan.
3) For the IF images in Figures 3, 6, & 7 can the author quantify the staining of multiple conidia to make the argue stronger, rather than just showing one (or a few) conidia) in each image.
4) Figure 5c: For the b1,3-glucan and a1,3-glucan stimulation it appears the authors stimulated with 1ug of each glucan, but is this equimolar? If not, how can the authors state one or the other drives increased amounts of cytokines. Secondly, what happens to CD40 and CD80 staining on the cells stimulated with these glucans, is it equivalent like the spores?
5) Figure 5: Overall, I would suggest show the data for each donor as a dot on each figure with an n=4, that seems small for a human study. Secondly, with an n=4 a non-parametric test like a Mann-Whitney U-test would be more appropriate. Both of these points are like important in Figure 5c for the analysis of the IL-8 secretion.
6) Figure 7: The IF images are not as sharp or bright as the previous images in Figures 2 & 6. Can the images be cleaned up or quantified like suggested in point 3 above. Moreover, the low levels of b1,3-glucan staining on the germinating spores doesn't match up with the earlier works looking at its role in Dectin 1 activation by Chad Steele (PMID: 16344862) and Tobias Hohl (PMID: 16304610).
Author Response
Reviewer 2
In this manuscript by Valsecchi et al they examine the immunostimulatory capacity of the ccrp mutants in the RodAp versus the complete rodA-null mutant strain of Aspergillus fumigatus. The authors find that the ccrp mutants are less immune stimulatory in terms of cytokines (IL-10 and IL-8), but induce similar levels of DC co-stimulatory molecules. The authors go on to try to demonstrate that this is due to the difference in alpha-1,3-glucan and beta-1,3-glucan exposure in the cells wall. This seems plausible but there are a couple controls that would strengthen their arguments and conclusions partially in regards to the glucan stimulations of the DCs. The biggest concern I have is the authors keep saying the ccrp-mutant does not have RodAp on the cell surface, but their proteome data argues against that.
Response: We agree with this reviewer, and we did not say that the RodAp was not found on the ccrp-mutant conidial surfaces. Instead, throughout the manuscript we mentioned that ccrp-mutations result in the conidia devoid of rodlets, an organized structure formed by the RodAp; it is the rodlet-structure that masks conidial cell wall PAMPs.
1) Section 3.2: I would argue based on the data from the proteomic analysis (Section 3.4) that the RodAp protein is in the cell wall (on the cell surface) somehow since you detect the presence of RodAp by LC-MS/MS, albeit about 5-fold reduced. I think this fact is very important for the interpretation of all the results especially the HF striping of the rodlets later in the manuscript, when those spores look more like the ccrp mutant than the rodA-null mutant.
Response: As responded before, we agree with this reviewer that the RodAp is present in the cell wall, evidenced by the LC-MS/MS analysis. What we pointed throughout the manuscript is that the ccrp-mutations have an impact on the folding of the RodAp, which in turn affect the stability, transport and organization of the RodAp into rodlet-structure.
2) Figure Legend 3: It appears the panel labels or the legend label are mixed up for which set of panels is b1,3-glucan vs. a1,3-glucan.
Response: We are sorry for this mistake, and it has been corrected.
3) For the IF images in Figures 3, 6, & 7 can the author quantify the staining of multiple conidia to make the argue stronger, rather than just showing one (or a few) conidia) in each image.
Response: In the revised manuscript, we have added Figure 3D and 6B, showing the flow cytometric analyses of the conidia labeled for b-1,3-glucan and a-1,3-glucan that confirms our argument of differential exposure of b-1,3-glucan and a-1,3-glucan on ccrp-mutant conidia compared to DrodA conidia. It will be difficult to perform such analysis for swollen conidia (Figure 7) as the swollen conidia aggregate. For clarity we have shown one (or a few) conidia; otherwise, we have repeated this experiment at least three-times, each time capturing five-eight images. Hope, this reviewer will be convinced by these results/our explanations.
4) Figure 5c: For the b1,3-glucan and a1,3-glucan stimulation it appears the authors stimulated with 1ug of each glucan, but is this equimolar? If not, how can the authors state one or the other drives increased amounts of cytokines. Secondly, what happens to CD40 and CD80 staining on the cells stimulated with these glucans, is it equivalent like the spores?
Response: β-(1,3)-Glucan and α-(1,3)-glucan vary in their chain lengths, and hence variable molecular weight. Therefore, we opted for absolute quantity rather than equimolar dose. The yield of α-(1,3)-glucan and β-(1,3)-glucan from one million conidia was nearly 1.1 and 3 μg respectively. As we used 1:1 ratio of conidia and DCs, we have selected 1 μg of each of the glucans in our study. We included these points in the revised text (Methods). The expressions (intensities) of CD40 and CD80 on dendritic cells stimulated with β-(1,3)-glucan and α-(1,3)-glucan were similar like those observed with spores. We have included the data in Figure 5C.
5) Figure 5: Overall, I would suggest show the data for each donor as a dot on each figure with an n=4, that seems small for a human study. Secondly, with an n=4 a non-parametric test like a Mann-Whitney U-test would be more appropriate. Both of these points are like important in Figure 5c for the analysis of the IL-8 secretion.
Response: In line with suggestions of the reviewer, we have included data from two additional donors (6 donors in total). We would like to mention here that our data are from independent donors who differ in genetic, epigenetic and environmental background. Statistical significance and uniform nature of the data thus validate our findings. We have now used Mann-Whitney U-test instead of Students t test (Fig 5C, D and E) when we are comparing only two groups. In other cases (Figure 5A, B) where we are comparing four groups, we used ANOVA (Dunnett’s post-test).
6) Figure 7: The IF images are not as sharp or bright as the previous images in Figures 2 & 6. Can the images be cleaned up or quantified like suggested in point 3 above? Moreover, the low levels of b1,3-glucan staining on the germinating spores doesn't match up with the earlier works looking at its role in Dectin 1 activation by Chad Steele (PMID: 16344862) and Tobias Hohl (PMID: 16304610).
Response: We tried to improvise the quality of the image in Figure 7; it is difficult to quantify the fluorescent intensity as during the isotropic swelling process conidia aggregate due to the exposure of alpha-1,3-glucan (PMID: 20447463; this reference has been included in the revised manuscript). In Hohl TM et al. (PMID: 16304610), A. fumigatus conidia germinated in RPMI for 3-7 h followed by heat killing (100OC for 30 min or autoclaving at 121OC for 15 min) have been used; heat inactivation will solubilize amorphous components leaving only fibrillar (beta-glucan and chitin) polysaccharides in the cell wall. In Steele C et al. (PMID: 16344862), Figure 5 indicates that 2 h swollen conidia show weak and punctate labeling by Dectin-1 (indeed many conidia were unlabeled) in agreement with our observation (Figure 7). In their study, an intense and uniform Dectin-1 labeling was observed only for 6 h swollen and 10 h germinating conidia.
Round 2
Reviewer 1 Report
The authors have adequately addressed my concerns.
Author Response
Thank you, and we have taken care of the language.
Reviewer 2 Report
Thank you for you thoughtful and thorough revision.
One minor point is in the flow cytometry histograms in Figure 3D, it is difficult to distinguish the groups, since you are already using color in the histogram maybe use all colors rather than dotted/dashed lines for some of the mutants. The data appear to complement the IF staining nicely, but it take a lot of work to figure that out as a reader.
Author Response
Thank you, and suggested by this reviewer, we have replaced histograms in Figure 3D with colored lines for clarity, as well taken care of the language.